# Theory of Planned Behavior applied to the choice of food with preservatives by owners and for their dogs

**Vivian Pedrinelli**[1], **Alexandre Rossi**[2], **Marcio A. Brunetto**[1] *

**1** School of Veterinary Medicine and Animal Science, University of Sao Paulo, Sao Paulo, Brazil, **2** Cão Cidadão, São Paulo, Brazil

* vivian.pedrinelli@gmail.com

**Data Availability Statement:** All relevant data are within the paper and its Supporting Information files.

**Funding:** The author(s) received no specific funding for this work.

## Abstract

Many pet owners make food choices for their pets that are similar to those they make for themselves, and food characteristics such as the presence of preservatives can influence this decision. The Theory of Planned Behavior (TPB) is a tool used to predict intentions and behavior and can be an important indicator for the pet food industry. The aim of this study was to investigate pet owner behavior regarding food with or without preservatives, based on the behavior prediction of TPB. A questionnaire was distributed with questions related to direct (attitude, subjective norms, and perceived behavioral control) and indirect (behavioral beliefs, normative beliefs, and intentions) measures for the analysis of TPB. For the statistical analysis the structural equation modeling (SEM) was used. The correlation between owner behavior and the behavior imposed on their dog's diet was evaluated by paired T test or paired Wilcoxon test according to variables' adherence or not to normality, respectively. A total of 1,021 answers were evaluated after the inclusion and exclusion criteria were applied. The results indicated that TPB was effective in predicting the intentions ($r^2 = 0.58$ for dogs and $r^2 = 0.59$ for owners) and behavior ($r^2 = 0.58$ for dogs and $r^2 = 0.57$ for owners) regarding the intake of diets without preservatives. It was observed that owners are more concerned with the diet of their dogs than their own and they believe that the intake of preservatives can be prejudicial to the health of their dogs ($p < 0.001$). However, owners trust more in pet food manufacturers than human food manufacturers ($p < 0.001$). Therefore, it can be concluded that TPB can be an important tool to understand consumer behavior towards their dog food, and that the industry should intensify its approach on safety of preservatives in pet food, since many owners still believe they can be prejudicial to dogs.

## Introduction

Dogs are increasingly considered as part of the family, and this humanization can lead owners to transfer their food choices to their pets [1–3]. To humans, the food choice is a complex act that involves social and cultural factors and can directly influence the choice of diet they make for their pets. Before buying a pet food, owners can take into consideration characteristics

**Competing interests:** The authors have declared that no competing interests exist.

such as quality, ingredients, animal preference, and flavor [4]. Several studies observed that the pet food characteristics, the sources from which the owners receive dietary recommendation, and the relationship between owners and their pets can influence the choice of food they will buy [4–6]. The difficulty in understanding labels can also influence this decision, since many owners do not understand ingredient description as well as the presence of preservatives and may opt for foods that do not contain these ingredients [2,7].

Food additives are ingredients added intentionally in diets with the aim of modifying the physical, chemical, biological, or sensory characteristics during the process of manufacturing a food [8]. Additives can be dyes, emulsifiers, or antioxidants, both natural such as tocopherols, or synthetic such as butylhydroxyanisole (BHA) and butylhydroxytoluene (BHT) [9]. Without these additives, there can be alterations in the food characteristics and, consequently, alterations in the nutritional value, including oxidation of fatty acids [9]. Despite the effect in food preservation, many owners are concerned about preservatives with allegations of toxicity or even carcinogenesis [2,10].

According to the Theory of Planned Behavior (TPB), the information alone does not produce or change behavior unless there is a change in the subject's attitude [11]. TPB proposes that behaviors can be predicted by: (a) attitudes; (b) subjective norms, which are how much the subject believes that others influence the behavior; (c) and perceived behavioral control, which is the perception of the subject over the ability to execute the behavior [11]. Few studies evaluated the beliefs of dog owners, and most of them show descriptive or quantitative results [4,10]. Therefore, behavioral analysis according to the TPB can bring important information to the pet food industry and professionals on how to approach some themes with pet owners [12].

The aim of this study was to investigate the applicability of TPB and its ability to predict dog owner behavior, as well as evaluate the difference in owner eating habits and the habits they impose on their dogs regarding foods with or without preservatives.

## Materials and methods

For this prospective study a questionnaire was elaborated containing two sets of questions: a first set with 16 questions related to the owners and dog characteristics; and a second set with 38 affirmations regarding direct and indirect measures for the TPB analysis. A pilot questionnaire was first applied and a convenience sample of 6 individuals was used to obtain feedback to elaborate the final version of the questionnaire. The final questionnaire was divided into four sections. The first section contained eight questions about the owners (i.e. age, gender, place of residence, scholarity, and monthly income). The second section was composed of eight questions about their dog (i.e. age, breed, weight, gender, how they acquired the dog, neutering, and type of food provided). The third section contained 19 affirmations in a 5-point Likert scale (Completely disagree to Completely agree) regarding the intentions, attitudes, behavioral beliefs, subjective norms, behavior, and perceived behavioral control applied to their choices of pet food for their dog. The fourth and final section of the questionnaire contained 19 questions similar to the third section, however focused on the owner's behavior towards the choice of their own food. The questions were adapted from previous studies on consumer behavior of pet owners and other individuals [6,12–14] (S1 File). In the case the owner had more than one dog, they were instructed to select only one dog and respond to the questionnaire according to that one dog selected.

An online tool was used (Google Forms, Google) for 30 days between January and February 2022, and the recruiting method was digital via social media. The questionnaire was elaborated in Portuguese to be answered in up to 15 minutes after consent to participate according to a

Free and Informed Consent Term at the beginning of the form. Only owners over 18 years of age that had at least one dog were included in the study. The exclusion criteria were owners under 18 years old, that did not have a dog, that did not consent to participate, and that were part of the pet food industry and/or were veterinarians or animal scientists.

Answers of the first and second sections of questions were evaluated descriptively. To evaluate the reliability between questions of each subgroup in the third and fourth sections of the questionnaire the Cronbach's α coefficient was used, and values above 0.7 were used to consider the sum of values of different affirmations for each measure (direct or indirect) to represent the constructs. Structural equation modeling (SEM) was used for the TPB analysis. To compare the answers of owner behavior and the behavior they apply to dog food, the Shapiro-Wilk test was used to evaluate the adherence to normality, and according to the results, either the paired T-test or the paired Wilcoxon test was used, considering values of $p < 0.05$ as significant. For the statistical analysis the softwares SPSS [15] and AMOS [16] were used.

Confirmatory Factor Analysis (CFA) was computed using AMOS (16) to test the measurement model. As part of the CFA, factor loadings were assessed for each item. The model-fit measurements were used to assess the model's fit (CFI, GFI, AGFI, TLI, SRMR, and RMSEA).

## Results

A total of 1,102 answers were obtained, from which 81 were excluded respecting the inclusion and exclusion criteria: one participant did not agree to participate, two were excluded for being under 18 years old, 20 were excluded for being part of the pet food industry, and 58 were excluded for being veterinarians or animal scientists. After applying these criteria, 1,021 questionnaires were considered for this study.

The responses for the first section, regarding the owner, were that 49.8% (n = 509/1021) lived in Sao Paulo state, 8.1% (n = 83/1021) lived in Minas Gerais state, and 7.4% (n = 76/1021) lived in Rio de Janeiro state in Brazil. Of all the participants, 57.6% (n = 588/1021) lived in houses, 41.2% (n = 421/1021) lived in apartments, and 1.2% (n = 12/1021) lived in small farms. Most participants were female (94.7% n = 967/1021), 5.0% (n = 51/1021) were male, and 0.3% (n = 3/1021) preferred not to declare. Other data regarding the owners are presented in Table 1.

Regarding the answers of section two, 59.2% (n = 604/1021) had only one dog, 32.9% (n = 336/1021) had 2 or 3 dogs, and 7.9% (n = 81/1021) had more than 3 dogs. The most common dog breeds were mixed breed (47.8%; n = 488/1021), Shih Tzu (7.8%; n = 80/1021), Yorkshire Terrier (5.7%; n = 58/1021), Lhasa Apso (3.2%; n = 33/1021) and Miniature Poodle and Pinscher, each with 2.1% (n = 21/1021). As for breed sizes, the majority were small breeds: 26.2% (n = 267/1021) had up to 6.5 kg, and 20.3% (n = 207/1021) had between 6.5 and 9 kg. Dogs between 9 and 15 kg consisted of 23.3% (n = 238/1021), dogs between 15 and 30 kg were 22.0% (n = 225/1021) and dogs over 30 kg were 8.3% (n = 84/1021). Most dogs were female (57.3%; n = 585/1021) and most of the dogs were neutered (68.3%; n = 697/1021). Regarding the relationship with the owner according to the area of the household the dogs were allowed to circulate in, 77.7% (n = 793/1021) of the animals had unrestricted access to rooms, 15.5% (n = 158/1021) had circulation restricted to some rooms, and 6.8% (n = 70/1021) were kept only outdoors. Information regarding the source of information on pet food and the type of food provided is presented in Table 2.

Table 3 presents the comparison between answers regarding the owner's dietary habits and the dietary habits they apply to their dog from sections three and four of the questionnaire. It was observed that owners are more or equally concerned about the diet of their dog than their own ($p < 0.001$) and believe that diets without preservatives are healthier ($p < 0.001$), despite

**Table 1. Results from section one regarding information about the dog owners.**

| Information | Option | Number of answers | % |
|---|---|---|---|
| Monthly income of household[†] | Up to US$ 1,184 | 410 | 40.2 |
| | From US$ 1,185 to US$ 2,369 | 259 | 25.4 |
| | More than US$ 2,369 | 166 | 16.2 |
| | Do not wish to disclose | 186 | 18.2 |
| Scholarity | No formal instruction or illiterate | 0 | - |
| | Did not complete middle school | 4 | 0.4 |
| | Completed middle school | 13 | 1.3 |
| | Did not complete high school | 12 | 1.2 |
| | Completed high school | 143 | 14.0 |
| | Did not complete graduate school | 126 | 12.3 |
| | Completed graduate school or post-graduation | 723 | 70.8 |
| Age | From 18 to 24 years | 43 | 4.2 |
| | From 25 to 34 years | 273 | 26.7 |
| | From 35 to 44 years | 341 | 33.4 |
| | From 45 to 54 years | 198 | 19.4 |
| | From 55 to 64 years | 139 | 13.6 |
| | 65 years or older | 27 | 2.6 |

[†]Values are showed in US dollars and are equivalent to 5 minimum salaries in Brazil (R$ 6,060) and 10 minimum salaries (R$ 12,120) at the time of the questionnaire, considering an exchange rate of 0.2 reais per US dollar.

not having an impact on consumer behavior (p = 0.538). Another information obtained is that owners trust more in pet food labels than labels for human products (p<0.001).

The results for Cronbach's α coefficient for the answers of sections three and four were all above 0.7, as presented in Table 4. Therefore, they were grouped for the SEM analysis.

The values used to assess the model's fit (CFI, GFI, AGFI, TLI, SRMR, and RMSEA) were within their acceptance levels for answers for dogs, except for RMSEA, which results in a reasonable fit for dogs and a good fit for owners (S1 and S2 Tables). The results of the SEM analysis [standard regression coefficients, which represent the correlation between variables, and the r-squared values ($r^2$), which represent how much the change in the variable can be explained by the other variables] are presented in Fig 1.

It can be observed that the subjective norms had a low correlation with intention but are more important for the diet of dogs (0.14) than owners (0.08). Attitude has more influence on

**Table 2. Results of questions from section two regarding source of information on pet food and type of food provided.**

| Information | Option | Number of answers | % |
|---|---|---|---|
| Source of information on pet food | My veterinarian or other professional | 637 | 62.4 |
| | Friends or family | 40 | 3.9 |
| | Social media | 117 | 11.5 |
| | Manufacturer website | 33 | 3.2 |
| | Other websites and blogs | 171 | 16.7 |
| | Other sources | 23 | 2.3 |
| Type of food provided as main meal | Dry kibble diet | 899 | 88.1 |
| | Wet diet | 17 | 1.7 |
| | Homecooked diet | 96 | 9.4 |
| | Raw meat-based diet | 9 | 0.9 |

**Table 3. Results of the comparison between answers of owners regarding their dietary behavior and the behavior they apply to their dog.**

| Measure | Negative classifications[1] (n) | Positive classifications[2] (n) | Ties[3] (n) | p |
|---|---|---|---|---|
| Attitude | 265 | 151 | 605 | <0.001 |
| Behavioral beliefs | 657 | 177 | 187 | <0.001 |
| Health | 394 | 192 | 435 | <0.001 |
| Trust in labels and products | 494 | 140 | 387 | <0.001 |
| Self-identity (concern about dietary habits) | 424 | 73 | 524 | <0.001 |
| Subjective norms | 159 | 335 | 527 | <0.001 |
| Normative beliefs | 265 | 398 | 358 | <0.001 |
| Influence of others | 204 | 370 | 447 | <0.001 |
| Personal influence | 201 | 183 | 637 | 0.597 |
| Perceived behavioral control | 236 | 320 | 465 | <0.001 |
| Intention | 343 | 233 | 445 | <0.001 |
| Behavior | 264 | 236 | 521 | 0.538 |

[1] Answers for which the owner chose a higher score on the Likert scale for the dog than for their own

[2] Answers for which the owner chose a lower score on the Likert scale for the dog than for their own

[3] Answers for which the owner chose the same score on the Likert scale for the dog and for their own.

the intention for diets of dogs (0.43) than owners (0.37). Regarding the $r^2$ values for diets of dogs, 58% of the change in intention can be explained by attitude, subjective norms, and perceived behavioral control, and the change in behavior can be explained by 58% of the perceived behavioral control and intention. In relation to diets for owners, 59% of the change in intention can be explained by attitude, subjective norms, and perceived behavioral control, and the change in behavior can be explained by 57% of the perceived behavioral control and intention.

## Discussion

Few studies used TPB to evaluate different aspects of owner behavior towards dogs. A study conducted by Rohlf et al. (2010) observed a correlation between the behavioral variables of the owners and their dog's body condition score (BCS), with the intention not being able to predict feeding habits but being able to predict exercises. The results of the present study showed that the TPB model was efficient in predicting the intention and owner behavior regarding

**Table 4. Results for the Cronbach's α coefficient for the questions from sections three and four of the questionnaire, according to the subgroups (if present).**

| Measure | Subgroup | Dog (section three) | | | Owner (section four) | | |
|---|---|---|---|---|---|---|---|
| | | Alpha[1] | Mean | SD | Alpha[1] | Mean | SD |
| Attitude | | 0.89 | 4.36 | 1.01 | 0.93 | 4.25 | 1.02 |
| Behavioral beliefs | Health | 0.76 | 3.65 | 1.15 | 0.75 | 3.27 | 1.15 |
| | Trust in labels and products | | | | | | |
| | Self-identity | | | | | | |
| Subjectve norms | | 1.00 | 3.22 | 1.19 | 1.00 | 3.58 | 1.20 |
| Normative beliefs | Influence of others | 0.82 | 3.48 | 1.20 | 0.85 | 3.61 | 1.13 |
| | Personal influence | | | | | | |
| Perceived behavioral control | | 0.84 | 3.51 | 1.23 | 0.87 | 3.59 | 1.21 |
| Intention | | 0.87 | 3.82 | 1.14 | 0.88 | 3.71 | 1.12 |
| Behavior | | 0.92 | 3.20 | 1.23 | 0.93 | 3.18 | 1.16 |

[1] Alpha = Cronbach's α coefficient calculated for the group of questions; SD = standard deviation.

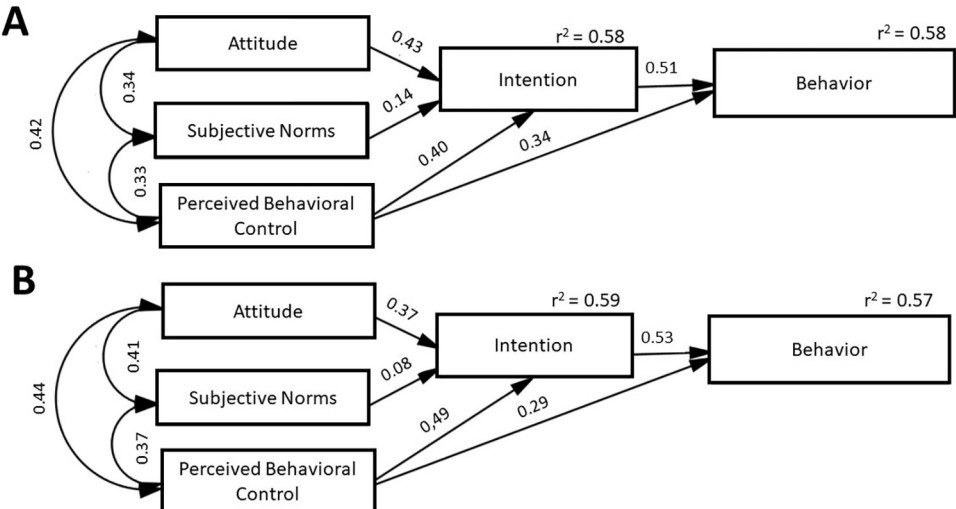

**Fig 1.** Theory of Planned Behavior (TPB) with the results of behavior toward consumption and purchase of food without preservatives for dogs (A) and owners (B). Values on arrows represent standardized regression coefficients.

foods without preservatives, which reinforces that TPB can be an interesting tool to better understand owner attitudes towards different aspects of their dog's life.

Another result observed in this study is that owners trust more in manufacturers of pet food than food intended for humans and that they are more concerned with their dog's diet than their own. In another study [17], it was identified, after applying a questionnaire, that owners had an increased tendency to buy healthier products for their dogs than for themselves, and that owners which have healthier eating habits are more likely to apply this behavior to their dog's diet. In another study, similar results were observed: 53.1% of participants referred to an equal priority of buying healthy foods for their dogs, and 43.6% reported a higher priority for healthier foods for their dogs [4].

Several studies observed that the presence of preservatives can be a deciding factor in the choice of diet for dogs. The absence of these additives is one of the main reasons for owners choosing raw diets for their pets [10]. Another study observed that 36.9% of owners don't trust the use of preservatives in pet food and 67.3% of owners consider the use of preservatives as a potential health risk for pets [18]. These previous results are similar to what was observed in the present study, in which the participants claimed that the intake of foods with preservatives can be a health risk for their dogs.

The knowledge regarding the use and importance of preservatives in pet food is a deciding factor for consumers of pet food. Shim et al. (2011) suggest that the knowledge of preservatives significantly increased the perception of the safety of their use. Therefore, it is recommended that professionals and pet food manufacturers bring correct information and awareness to owners regarding the purpose and safety of additives, especially preservatives, in pet food.

Despite the owner's distrust in the safety of preservatives in pet food, few studies referred to the side effects of these additives [19]. BHA, one of the most common synthetic preservatives used in dog food, has a current maximum recommended inclusion of 150 mg/kg diet, alone or when used with BHT [20]. Some of the side effects include lower albumin concentrations and increased concentrations of alkaline phosphatase in dogs fed over 5,000 mg/kg diet of BHA, which is more than 33 times the maximum recommended BHA levels in pet food [21]. In adult dogs fed diets containing over 10,000 mg BHA/kg diet, which is more than 68 times the current limit for BHA, histological liver alterations were observed [22]. Preservatives, however,

are not only used in dry kibble diets. Sulfites can be used as antioxidants in meats, and in excess were associated with clinical signs of thiamine deficiency in dogs [23]. Therefore, it is important to consider that preservatives, as well as essential nutrients such as cholecalciferol or retinol, can be beneficial when used in the adequate amount and prejudicial when used in excess [19,24,25]. Being aware of this information will favor the purchase of products from manufacturers that respect the legal recommendations and supply products that are safe and complete.

Limitations include the method being a questionnaire, and therefore an involuntary bias could have occurred due to owners answering what they think is correct rather than what reflects their reality. The authors attempted to mitigate this by running a pilot of the questionnaire and adjusting questions as necessary before running the actual research. The number of answers could also have reduced the effects of bias, since people from all over the country and from different social and financial realities were impacted by the questionnaire, which ensured a diversity of respondents.

It can be concluded that the TPB can be an interesting tool to aid the pet food industry and professionals to better understand consumer behavior towards dogs, which helps in approaching subjects that can bring doubt, such as the use of food preservatives. As we observed, subjective norms and attitude have more influence in the intention on buying foods without preservatives for dogs that foods for the owner themselves, which could mean that owners are more concerned with the impact of their dog diet on their pet's health than the impact of their own diet on their health. As many owners believe that preservatives are harmful to their pets, it is suggested that more information on the safety of their use is brought to light.

## Supporting information

**S1 File. Questionnaire.**
(DOCX)

**S1 Table. Factor loadings.**
(DOCX)

**S2 Table. Model fit.**
(DOCX)

## Acknowledgments

Marcio A. Brunetto passed away before the submission of the final version of this manuscript. Vivian Pedrinelli accepts responsibility for the integrity and validity of the data collected and analyzed.

## Author Contributions

**Conceptualization:** Vivian Pedrinelli.

**Formal analysis:** Vivian Pedrinelli.

**Investigation:** Alexandre Rossi.

**Methodology:** Vivian Pedrinelli, Marcio A. Brunetto.

**Writing – original draft:** Vivian Pedrinelli.

**Writing – review & editing:** Marcio A. Brunetto.

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
