## [Decision Letter · Decision Letter 0]

30 Mar 2023

PONE-D-23-03192Theory of Planned Behavior applied to the choice of food with preservatives by owners and for their dogsPLOS ONE

Dear Dr. Pedrinelli

Thank you for submitting your manuscript to PLOS ONE. After careful consideration, we feel that it has merit but does not fully meet PLOS ONE’s publication criteria as it currently stands. Therefore, we invite you to submit a revised version of the manuscript that addresses the points raised during the review process.

ACADEMIC EDITOR:   Authors are advised to test for the existence of common method bias in the dataset given that there is high probability of receiving socially desirably answers from the respondents. Provide all the relevant results from the analysis including the measurement model results, the model fitness and the structural model results. Highlight the implications of the key findings in the concluding part of the paper. I wish you all the best

We look forward to receiving your revised manuscript.

Kind regards,

Godfred Matthew Yaw Owusu

Academic Editor

PLOS ONE

Additional Editor Comments :

Authors are advised to test for the existence of common method bias in the dataset given that there is high probability of receiving socially desirably answers from the respondents. Provide all the relevant results from the analysis including the measurement model results, the model fitness and the structural model results. Highlight the implications of the key findings in the concluding part of the paper. I wish you all the best

Sincerely,

Prof. Godfred Matthew Yaw Owusu

Reviewer(s)' and Academic Editor Comments to Author

Reviewer 1

The authors exhibit a strong methodological approach which ensures the fitness of the model and reliability of the data used for the study. The source of the scales for measuring the variables used are provided; diagnostic tests such as normality and validity tests were conducted. The authors portray a good understanding of the subject area, existing literature and the Theory of Planned Behaviour employed in explaining their findings. Further, the paper is written in an easy to read way which enhances understandability.

Given that the data was collected using google forms via social media, there is the risk of reaching the wrong audience which could affect the data. Again, using self-reported questionnaires poses the risk of receiving socially desirable answers. However, the authors did not indicate how this risk is mitigated.

The authors should include a section that highlights the various limitations of the study, especially with the data collection process, and how these were addressed.

Reviewers' comments:

Reviewer's Responses to Questions

**Comments to the Author**

1. Is the manuscript technically sound, and do the data support the conclusions?

Reviewer #1: Yes

2. Has the statistical analysis been performed appropriately and rigorously? 

Reviewer #1: Yes

3. Have the authors made all data underlying the findings in their manuscript fully available?

Reviewer #1: Yes

4. Is the manuscript presented in an intelligible fashion and written in standard English?

Reviewer #1: Yes

5. Review Comments to the Author

Reviewer #1: The authors exhibit a strong methodological approach which ensures the fitness of the model and reliability of the data used for the study. The source of the scales for measuring the variables used are provided; diagnostic tests such as normality and validity tests were conducted. The authors portray a good understanding of the subject area, existing literature and the Theory of Planned Behaviour employed in explaining their findings. Further, the paper is written in an easy to read way which enhances understandability.

Given that the data was collected using google forms via social media, there is the risk of reaching the wrong audience which could faulter the data. Again, using self-reported questionnaires poses the risk of receiving socially desirable answers. However, the authors did not indicate how this risk is mitigated.

The authors should include a section that highlights the various limitations of the study, especially with the data collection process, and how these were addressed.

6. PLOS authors have the option to publish the peer review history of their article (what does this mean?). If published, this will include your full peer review and any attached files.

Reviewer #1: No

---

## [Author Response · Author response to Decision Letter 0]

14 Aug 2023

ACADEMIC EDITOR: 

 Authors are advised to test for the existence of common method bias in the dataset given that there is high probability of receiving socially desirably answers from the respondents. Provide all the relevant results from the analysis including the measurement model results, the model fitness and the structural model results. Highlight the implications of the key findings in the concluding part of the paper. I wish you all the best.

Response: Thank you for your comment. We added wo tables in the Supplement section (S2) that include the model fit and factor loadings. We added a description in the Material and Methods section to describe the CFA (lines 135-138) as well as in the Results section (lines 204-207). We also added in the conclusion the highlights of the findings (lines 287-291).

Reviewer 1

The authors exhibit a strong methodological approach which ensures the fitness of the model and reliability of the data used for the study. The source of the scales for measuring the variables used are provided; diagnostic tests such as normality and validity tests were conducted. The authors portray a good understanding of the subject area, existing literature and the Theory of Planned Behaviour employed in explaining their findings. Further, the paper is written in an easy to read way which enhances understandability.

Given that the data was collected using google forms via social media, there is the risk of reaching the wrong audience which could affect the data. Again, using self-reported questionnaires poses the risk of receiving socially desirable answers. However, the authors did not indicate how this risk is mitigated.

The authors should include a section that highlights the various limitations of the study, especially with the data collection process, and how these were addressed.

Response: Thank you for your comments. We added a paragraph with limitations of the study and how they were addressed (lines 276-283).

---

## [Decision Letter · Decision Letter 1]

25 Oct 2023

Theory of Planned Behavior applied to the choice of food with preservatives by owners and for their dogs

PONE-D-23-03192R1

Dear Dr. Pedrinelli,

We’re pleased to inform you that your manuscript has been judged scientifically suitable for publication and will be formally accepted for publication once it meets all outstanding technical requirements.

Kind regards,

Godfred Matthew Yaw Owusu

Academic Editor

PLOS ONE

Additional Editor Comments (optional):

Thanks for addressing the comments by the reviewers.

Reviewers' comments:

Reviewer's Responses to Questions

**Comments to the Author**

1. If the authors have adequately addressed your comments raised in a previous round of review and you feel that this manuscript is now acceptable for publication, you may indicate that here to bypass the “Comments to the Author” section, enter your conflict of interest statement in the “Confidential to Editor” section, and submit your "Accept" recommendation.

Reviewer #1: (No Response)

2. Is the manuscript technically sound, and do the data support the conclusions?

Reviewer #1: (No Response)

3. Has the statistical analysis been performed appropriately and rigorously? 

Reviewer #1: (No Response)

4. Have the authors made all data underlying the findings in their manuscript fully available?

Reviewer #1: (No Response)

5. Is the manuscript presented in an intelligible fashion and written in standard English?

Reviewer #1: (No Response)

6. Review Comments to the Author

Reviewer #1: (No Response)

7. PLOS authors have the option to publish the peer review history of their article (what does this mean?). If published, this will include your full peer review and any attached files.

Reviewer #1: No

---

## [Editor Report · Acceptance letter]

9 Jan 2024

PONE-D-23-03192R1 

PLOS ONE

Dear Dr. Pedrinelli, 

I'm pleased to inform you that your manuscript has been deemed suitable for publication in PLOS ONE. Congratulations! Your manuscript is now being handed over to our production team.

Kind regards, 

on behalf of

Dr. Godfred Matthew Yaw Owusu 

Academic Editor

PLOS ONE